# Identification of a Rice Leaf Width Gene *Narrow Leaf 22* (*NAL22*) through Genome-Wide Association Study and Gene Editing Technology

**DOI:** 10.3390/ijms24044073

**Published:** 2023-02-17

**Authors:** Yuchen Xu, Shuangyong Yan, Su Jiang, Lu Bai, Yanchen Liu, Shasha Peng, Rubin Chen, Qi Liu, Yinghui Xiao, Houxiang Kang

**Affiliations:** 1College of Agronomy, Hunan Agricultural University, Changsha 410128, China; 2State Key Laboratory for Biology of Plant Diseases and Insect Pests, Institute of Plant Protection, Chinese Academy of Agricultural Sciences, Beijing 100193, China; 3Tianjin Key Laboratory of Crop Genetic Breeding, Tianjin Crop Research Institute, Tianjin Academy of Agriculture Sciences, Tianjin 300112, China

**Keywords:** Cell division, CRISPR/Cas9 gene editing, Genetic architecture of rice leaf width, Genome-wide association study, Gibberellin, Rice leaf width, Vein width

## Abstract

Rice leaf width (RLW) is a crucial determinant of photosynthetic area. Despite the discovery of several genes controlling RLW, the underlying genetic architecture remains unclear. In order to better understand RLW, this study conducted a genome-wide association study (GWAS) on 351 accessions from the rice diversity population II (RDP-II). The results revealed 12 loci associated with leaf width (LALW). In LALW4, we identified one gene, *Narrow Leaf 22* (*NAL22*), whose polymorphisms and expression levels were associated with RLW variation. Knocking out this gene in Zhonghua11, using CRISPR/Cas9 gene editing technology, resulted in a short and narrow leaf phenotype. However, seed width remained unchanged. Additionally, we discovered that the vein width and expression levels of genes associated with cell division were suppressed in *nal22* mutants. Gibberellin (GA) was also found to negatively regulate *NAL22* expression and impact RLW. In summary, we dissected the genetic architecture of RLW and identified a gene, *NAL22*, which provides new loci for further RLW studies and a target gene for leaf shape design in modern rice breeding.

## 1. Introduction

The leaf is an important organ that synthesizes carbohydrates from photosynthesis [1]. The size, shape, and color of leaves directly affects the efficiency of light utilization [2,3]. Rice leaf architecture is crucial in rice production, but improving it remains challenging. During the reproductive stage, the top three leaves are referred to as “functional leaves” as they contribute to most of the energy of rice seeds [4]. Studies have found that over 80% of carbohydrates in rice grains come from the first (flag leaf) and the second leaves [4,5]. According to Yuan Longping’s ideal plant type for super hybrid rice [6], the functional leaves should be “long, straight, narrow, concave and thick”. A narrow leaf phenotype can improve the rice morphological characteristics and promote an upright and compact plant type, which reduces shading and increases the photosynthetic area, ultimately improving the plant’s overall performance. Therefore, identifying the genes that control RLW is essential.

Recently, several narrow-leaf mutants have been generated through methods such as EMS mutagenesis [7], radiation mutagenesis [8], and T-DNA insertion [9], leading to the identification and cloning of several narrow-leaf genes. These genes regulate leaf width (LW) through various pathways, including affecting the vascular pattern and regulating cell division or elongation. For instance, *NAL1* [10,11,12], which encodes an unknown function protein, regulates the leaf blades’ width by modulating the vascular pattern and cell elongation, as well as affecting the polar transport of auxin. *NAL2* and *NAL3* [13], which encode the WUSCHEL-related homeobox 3A transcription factors, function redundantly, as only the double mutant, *nal2*-*nal3*, exhibited a narrow leaf phenotype. Overexpression of *NAL2* and *NAL3* results in a dwarf and wide-leaf phenotype due to the inhibition of GA_1_ biosynthesis [14]. *NAL21* [15] is another gene associated with leaf width, as the *nal21* mutant showed a narrow leaf phenotype with fewer large and small veins and a smaller number and width of cells along the leaf-width direction. Other genes, such as *AVB* [16], *CCC1* [17], and *DNL4* [18] are also related to leaf width. Although a number of genes have been identified, the genetic architecture of RLW remains unclear.

GWAS is an efficient approach for mapping agronomic trait genes. Over the past two decades, numerous loci and genes associated with rice agronomic traits, such as plant height and tiller number have been identified through GWAS [19]. In the case of rice leaves, using 183 landrace rice varieties, Roy [20] performed GWAS and identified 20 loci associated with leaf width. Meng [21] also found three quantitative trait loci associated with flag leaf width. However, no RLW gene has yet been cloned using the GWAS strategy.

In this study, we conducted a field experiment with 351 rice varieties and measured the RLW phenotype. Then, we performed a GWAS and identified a leaf-width-associated gene, *Narrow Leaf 22* (*NAL22*), on chromosome 3. The CRISPR-Cas9 knockout mutant *nal22* had narrower leaves than the wild type (WT). Anatomical observation revealed that the large and small veins were narrower in the *nal22* mutant. Furthermore, qRT-PCR results showed that the expression levels of cell division-related marker genes were down-regulated in the *nal22* mutant. This study identified a leaf-width-related gene through GWAS and gene-editing technology and provided a method for RLW gene cloning and resource for leaf shape design in modern rice breeding.

## 2. Results

### 2.1. Leaf Width Variations in the RDP-II Population

To dissect the genetic architecture of rice leaf width, we selected 351 rice accessions consisting of five subpopulations: Indica (IND), Temperate Japonica (TEJ), Tropical Japonica (TRJ), Aromatic (ARO), and aus (AUS), from RDP-II as experimental materials. These accessions were planted in a field in Tianjin, and the second leaf’s width was measured in the adult stage (Appendix A). The leaf width of 351 accessions was found to be close to a normal distribution (Figure 1A), with an average leaf width of 1.63 ± 0.28 cm. The widest and narrowest leaves were 3.5 cm (accession: 117444) and 0.9 cm (accession: 120957), respectively. Over 56% (197/351) of the leaves had widths between 1.4 and 1.8 cm, with only six accessions having widths less than 1 cm or greater than 2.2 cm (Figure 1A). The leaf width varied among different subpopulations, with TRJ having the largest average leaf width, followed by TEJ, AUS, IND, and ARO (Figure 1B). Among them, the differences between TRJ and IND/ARO were significant.

### 2.2. Identification of LALWs and Candidate Genes in the Rice Genome

Using the leaf-width phenotype measured in the field and high-density single nucleotide polymorphism (SNP) maps (131,332 SNPs with MAF ≥ 0.05) of RDP-II, we performed a GWAS and identified 12 LALWs (Figure 1C and Appendix A). These 12 LALWs were located on rice chromosomes 1, 3, 4, 5, 6, 7, 9, and 10.

When comparing the 12 LALWs to the 105 known leaf width-associated loci (Appendix A), we found that LALW4 was colocalized with FLW3, and LALW5 was colocalized with FW4-1 and FLW4-1 (Appendix A). These two loci were identified through different mapping populations and methods, suggesting that they are more stable and likely to contain genes controlling leaf width. However, no leaf-width genes have been cloned in these loci. Thus, we analyzed the 500 kb genomic sequence (250 kb flanking the top associated SNPs) of LALW4 and LALW5 and found 150 predicted genes according to the genome annotations of Nipponbare (NPB) (Appendix A). Among them, the numbers of transposons, retrotransposons, expressed proteins, and hypothetical proteins were 10, 8, 47, and 3, respectively. The other 82 genes were annotated with putative functions. Seven of these genes were either associated with plant hormone pathways or highly expressed in rice leaves, making them potential candidate genes (*CG*s) for affecting leaf width. *CG1* (LOC_Os03g50860) encodes a histidine kinase, a member of the cytokinin receptor kinase family, which can recognize trans-zeatin [22]. *CG2* (LOC_Os03g51030) encodes phytochrome A, which decreases the content of active gibberellin by inhibiting the expression of the GA biosynthesis genes *GA3ox2* [23] and *GA20ox* [24]. *CG3* (LOC_Os04g44230) encodes a cytokinin dehydrogenase precursor. *CG4* (LOC_Os04g44150) encodes GA2ox6, which belongs to the GA2-oxidase family and inactivates bioactive GA. Overexpression of *GA2ox6* could reduce the content of active gibberellin and lead to a semi-dwarf phenotype [25]. *CG5* (LOC_Os04g44780) encodes a beta-expansin precursor protein that belongs to the expansin family, which can promote cell wall extension. *EXPB2* [26], a homolog gene of *CG5*, has been proven to impact leaf width. Then, the expression patterns of the 82 genes in the two target loci were analyzed using an online tool (http://bioinfo.sibs.ac.cn/plantregulomics/index.php, accessed on 28 December 2021). We obtained 77 gene expression patterns among different rice tissues (Appendix A). Given that the leaf-width-associated genes are related to leaf development, we first screened genes whose FPKM > 4 in the 20-day-old leaves. In total, 62.3% (48/77) of the genes met the criteria. These genes were further analyzed for their expression levels in different tissues. Some were highly expressed in leaves but not in reproductive organs or seeds, suggesting they play important roles in leaf development. Finally, we identified another two candidate genes, LOC_Os04g44200 (*CG6*) and LOC_Os03g51479 (*CG7*). *CG6* is highly expressed in the leaf, pistil, and lamina joints of the flag leaf and poorly expressed in the root, embryo, endosperm, and seed. It is annotated as oxygen-evolving enhancer protein 3, a chloroplast precursor involved in photosynthesis. CG7 is highly expressed in the leaf, lamina joints of the flag leaf, and root but not in reproductive tissues (Appendix A).

### 2.3. Sequence Polymorphisms of CG7 Were Associated with Leaf Width

To explore the relationship between the seven candidate genes and leaf-width phenotype, we randomly selected 6 wide-leaf (>2 cm) and 6 narrow-leaf (<1.3 cm) accessions for genotyping. Using gene-specific primers, we amplified the seven genes’ DNA sequences from about 2000 bp upstream of the 5′-untranslated region (5′-UTR) to 1000 bp downstream of the 3′-UTR. Sequence analysis results showed that there were 16, 17, 6, 3, 6, 2, and 15 SNPs/insertions/deletions polymorphisms among the selected 12 accessions for *CG1-CG7*, respectively (Figure 2 and Appendix A). Of the 65 polymorphisms, 50.8% (33) were distributed in upstream regions, 27.7% (18) in downstream regions, and 16.9% (11) in introns. Only 4.6% (3) were in exons, with one causing a missense mutation. In *CG1-CG6,* none of these polymorphisms were closely associated with the leaf-width phenotype (Appendix A). However, in *CG7*, we identified 15 variations associated with the leaf-width phenotype (Figure 2). Among them, 10 SNPs and one insertion were 100% associated with leaf width. Three, two, and four SNPs were located in the promoter (2020, 1387, 1103 bp upstream of the start codon), intron (248 and 1090 bp downstream of the start codon), and downstream (183, 1230, 1345, and 1545 bp downstream of the stop codon) regions, respectively. The insertion was also located in the downstream region (Figure 2).

### 2.4. The Transcription Level of CG7 Was Associated with Leaf Width

As mentioned above, none of the leaf-width-associated variations in *CG7* were located in the exonic region. It is speculated that these variations might affect the transcription level of *CG7*. To verify this speculation, the transcription level of *CG7* was tested in rice accessions with extreme leaf widths. The results showed that, on average, the transcription level of *CG7* was higher in wide-leaf accessions than in narrow-leaf accessions (*p* = 0.0004) (Figure 3). This result demonstrated that the leaf-width phenotype was positively associated with the transcription level of *CG7*. 

### 2.5. CG7 Regulates Leaf Development

To determine the role of *CG7* in regulating leaf development, *CG7* knockout mutants was created on a Zhonghua11 (ZH11) background using CRISPR/Cas9 gene editing technology. The deletion of ‘GG’ and ‘ACTG’ in the target led to a premature stop codon in the T1 generation transgenic plants (Figure 4A). Two homozygous lines, cg7-2 and cg7-3, were selected for subsequent experiments. Compared with WT, the second leaves of these two mutants in the heading stage were significantly narrower (*p* = 6.99 × 10^−7^ for *cg7-2* and 3.62 × 10^−6^ for *cg7-3*) and shorter (*p* = 2.67 × 10^−10^ for *cg7-2* and 5.35 × 10^−10^ for *cg7-3*). The mean leaf width of the wild type was 1.201 ± 0.03 cm, while the mean leaf widths of cg7-2 and cg7-3 were only 1.041 ± 0.05 cm and 1.084 ± 0.05 cm, respectively (Figure 4B,C). The average leaf length of the WT was 55.2 ± 0.79 cm, and that of the two mutants were 44.69 ± 1.35 cm and 47.78 ± 0.92 cm (Figure 4B,D). The above results proved that *CG7* plays an important role in regulating leaf width and length. Therefore, we named *CG7 Narrow Leaf 22* (*NAL22*).

A previous study showed a correlation between leaf width and seed width in different rice varieties [27]. Therefore, we first examined whether there was a similar correlation in RDP-II. The seed widths of 12 accessions used for gene sequencing were measured under a microscope. Four of the six wide-leaf accessions had wider seeds, whereas four of the six narrow-leaf accessions had narrower seeds (Figure 5A,B). However, the seed widths of the *nal22* mutants were measured and we found no significant difference compared with the WT (Figure 5C,D). The average seed widths of the WT and *nal22* mutants were 2.98 ± 0.06 mm, 2.98 ± 0.09 mm, and 2.99 ± 0.05 mm, respectively. This suggested that *NAL22* regulates leaf width but not seed width.

### 2.6. NAL22 Regulates the Width of Veins

Midveins, large veins, and small veins were major longitudinal veins (Figure 6A). Previous anatomical studies demonstrated that the number and width of veins contribute to leaf width [13]. We compared the veins of the WT and mutants (*nal22*). There was no significant change in the total number of large and small veins between the WT and *nal22* leaves (Figure 6B and Appendix A). Additionally, the number of small veins between two large veins was the same in *nal22* as in the WT (Appendix A). Notably, the same number of veins in *nal22* resulted in narrower leaves, indicating that the density of veins, not the number, led to the narrower leaf phenotype in *nal22* (Figure 6B). The width of large and small veins was also measured and compared. The results showed that both large and small veins in *nal22*-2 and *nal22*-3 mutants were narrower (*p* = 7.92 × 10^−7^ for large veins and 0.0017 for small veins in *nal22-2*, *p* = 2.56 × 10^−6^ for large veins and 6.26 × 10^−5^ for small veins in *nal22-3*) compared with those in the WT. The average width was 150.1 ± 5.42 μm for the large veins and 47 ± 5.20 μm for the small veins in the WT leaves. However, the mean widths of the large veins in the *nal22*-2 and *nal22*-3 mutants were only 124.9 ± 7.56 μm and 120.2 ± 8.42 μm, respectively (Figure 6C). The mean widths of the small veins were 35.0 ± 5.23 μm and 31 ± 4.03 μm in the two *nal22* mutants, respectively (Figure 6D). The above results indicated that *NAL22* was associated with leaf width due to its positive influence on vein width.

### 2.7. NAL22 Regulates Cell Division

The other key factors impacting leaf width are cell division and cell expansion. Cell division affects the number of cells, while cell expansion determines the size of cells. Therefore, we did qRT-PCR to analyze the expression levels of genes involved in cell division and cell expansion in WT and *nal22*. The results showed that the expression levels of the mitotic marker genes, *CDC20S-1* (G1 and G2 period), *CDC20S-2* (G1 and G2 period), and *R2* (S period), were decreased by approximately half in *nal22* compared to the WT (Figure 6E–G). A critical factor in cell expansion is cell wall loosening, and expansins and GH9 enzymes are two types of proteins involved in this process. The expression levels of three cell wall loosening-associated marker genes, including *EXPA10*, *EXPB5* and *GH9B1*, were not significantly different between the WT and *nal22* mutants (Appendix A). Taken together, these results indicated that *NAL22* plays an important role in regulating cell division rather than cell expansion.

### 2.8. NAL22 Expressed Highly in Roots and Leaves

Based on the close relationship between gene expression and their functions [28], the temporal and spatial expression patterns of *NAL22* in rice tissues were examined. Total RNA was extracted from the roots, stems, and leaves of ZH11 at the seedling, tillering and filling stages. qRT-PCR results showed that *NAL22* was present in all tested parts of the plant but displayed different expression levels across growth stages (Figure 7A). *NAL22* was relatively highly expressed in leaves in the seedling and filling stages, and in roots during the tillering stage. We further measured the root length of two-week-old seedlings, where the root length of *nal22* was significantly shorter than that of the wild-type (*p* = 0.0236 for *nal22-2*, *p* = 0.0173 for *nal22-3*) (Appendix A). These results suggested that *NAL22* may play an important role in leaf and root development.

### 2.9. NAL22 Encoded a Conserved Plant Protein

*NAL22* encodes a 322 amino acid protein belonging to the Maf-like nucleoside triphosphate pyrophosphatase protein family and contains two maf domains. To explore whether it was conserved in plants, the NAL22 protein sequence was used as a query to search for orthologs, resulting in the discovery of orthologs in 78 plant species, including 56 dicotyledons and 22 monocotyledons. This demonstrates that NAL22 was conserved across plants. A phylogenetic analysis was then conducted on NAL22 and its orthologs, and it found that the genetic relationship was closely related to the differentiation of monocotyledon and monocotyledon (Appendix A). NAL22 exhibited the highest similarities with orthologs named maf-like protein (MafL), 7-methyl-GTP pyrophosphatase-like protein (m7GTP), and hypothetical protein (HyPr) in *Sorghum*, *Panicum virgatum* and *Oryza meyeriana*, respectively. Additionally, all members of the Gramineae family were clustered in a major clade, suggesting that NAL22 may have conserved functions. Further protein motif analysis revealed five conserved motifs, three of which were present in all 79 proteins, indicating their crucial role in NAL22 function.

### 2.10. NAL22 Was Negatively Regulated by GA

In plant biology, several leaf-width genes are associated with plant phytohormones, such as *GRF1* [28], *NAL2/3* [29], and *OFP2* [30]. The question was raised whether phytohormones also regulated *NAL22*. By analyzing the *cis*-acting elements of *NAL22*, several motifs were identified (Appendix A), with a focus on a TATC-box, which was found to be a GA response element [31,32]. It prompted us to detect whether *NAL22* was also related to GA. GID1 is the receptor of GA; a mutation of *GID1* in rice obstructs GA signal transduction, resulting in dwarfism. The leaf width of *gid1* mutant was significantly wider than that of NJ6 (wild type of *gid1*), with an average width of 1.55 ± 0.06 cm and 0.93 ± 0.08 cm (Figure 7B,C), indicating a negative impact of GA on leaf width. To determine whether *NAL22* expression was affected by GA, the expression level of *NAL22* was detected in two-week-old rice seedlings of NJ6 and *gid1* using qRT-PCR. The result showed that *NAL22* was up-regulated by almost 6 times in the *gid1* mutant compared to NJ6 (Figure 7D). To further test this hypothesis that GA negatively regulates the expression level of *NAL22*, NJ6 was treated with water, GA_3_ (a type of bioactive GA), and PAC (a GA biosynthesis inhibitor) and the leaves were collected at different time points after treatment. qRT-PCR results showed that *NAL22* was significantly down-regulated in response to GA_3_ treatment but up-regulated in response to PAC treatment (Figure 7E). It was confirmed that *NAL22* was regulated by GA. At the same time, the treatment of GA_3_ did not alter the expression of *NAL22* in the *gid1* mutant (Appendix A), indicating that *NAL22* function downstream of *GID1*.

## 3. Discussion

### 3.1. GWAS for Leaf Width

In rice, numerous loci and genes associated with different agronomic traits have been identified through GWAS [33], such as *OsSPL13* [34], *OsGSK3* [35], and *OsGSE5* [36]. However, research on mapping the leaf-width phenotype is still limited. In this study, a GWAS was conducted using 351 rice accessions from the RDP-II and 131,332 SNPs to identify the loci associated with leaf width (Figure 1). Twelve loci were mapped and one leaf-width-associated gene was identified (Figure 2). Knocking out this gene resulted in a narrow leaf phenotype (Figure 3B). Therefore, we provide a method combining GWAS and gene-editing technology to clone the rice leaf-width gene.

### 3.2. NAL22 Regulates Leaf Width

Approximately 50 leaf -width-related genes have been reported. These genes can be classified into three types based on their functions in determining. The first type of gene regulates leaf width by affecting the vein pattern. *NAL1*, *NAL2/3*, *NAL21*, and *NRL1* influence leaf width by controlling the number of veins [13,15,37,38]. In mutants of the above genes, the number of veins is reduced, resulting in narrow leaves. In the *mr11* narrow leaf mutant, the number of veins and spacing between small veins are decreased [39]. In this study, the *nal22* mutant showed a narrow leaf phenotype, but anatomical observations revealed that the number of large and small veins, as well as the number of small veins between adjacent large veins, did not decrease (Appendix A). Interestingly, the *nal22* mutant showed significantly smaller vein width in both large and small veins compared to the wild type (Figure 4C,D). The width of veins is mainly determined by the xylem vessel, which positively correlates with stem thickness. Meng [21] found a significant positive correlation between stem thickness and leaf width, demonstrating the impact of xylem vessels on the width of veins. Our findings also indicated that the width of veins is important for determining leaf width.

The second type of gene affects leaf width through cell division or elongation regulation. Leaf size is primarily determined by the number and size of leaf cells [15]. Disrupting the *AVB* and *CD1* or over-expressing *ARF19* results in a narrow-leaf phenotype, which partially depends on the reduced cell number [9,16,40]. Additionally, *CCC1*, *EXPB2*, and *SNFL1* regulate leaf size by controlling cell elongation [17,26,41]. Other genes, such as *DWARF* and *NAL21*, have been found to have dual effects on leaf size [15,42]. In this study, the transcription levels of genes involved in cell division or elongation were examined. The results showed that *CDC20S-1*, *CDC20S-2*, and *R2*, related to cell division, were significantly down-regulated in *nal22* mutants (Figure 4E–G). However, compared with the wild type, the transcription levels of cell elongation genes were not significantly changed in *nal22* mutants (Appendix A). These results suggest that the decreased leaf width in the *nal22* mutant is due to the inhibition of cell division.

The third type of gene influences leaf width through its impact on plant phytohormones, including biosynthesis and signal transduction. Auxin (IAA), cytokinin (CTK), brassinosteroids (BR), and GA have all been reported to play roles in regulating rice leaf width. *NAL1* regulates leaf width by affecting auxin polar transport [37]. Knockdown of *WOX4* decreased the expression of two cytokinin synthesis genes, leading to a reduction in cytokinin content and a subsequent impact on leaf development [43]. Overexpression of *SG1* in rice results in wide, dark-green, and erect leaf phenotypes similar to those seen in BR-deficient mutants [44]. However, overexpression of *SG1* did not alter BR synthesis but weakened the response. *GID2* encodes an E3 ubiquitin ligase that mediates the degradation of the GA pathway repressor DELLA protein. In the *gid2* mutant, GA signal transduction is largely blocked, resulting in wider leaves than in the WT [45]. In *slr1* (DELLA protein in rice) mutants, leaves were narrower than in the WT [46]. Corresponding with the phenotype of the *gid2* mutant, *gid1* also had a wider leaf than in the WT (Figure 7D), suggesting that GA negatively regulated leaf width. In this study, we found a GA response element, TATC-box, on the promoter of *NAL22* (Appendix A). Therefore, *NAL22*’s involvement in the GA pathway was further studied. The transcription level of *NAL22* was found to be significantly higher in the *gid1* mutant than in the WT (Figure 5B). In GA_3_-treated WT, the transcription level of *NAL22* decreased continuously within 24 h, while PAC treatment significantly increased *NAL22* expression (Figure 5C). There was no significant change in the transcription level of *NAL22* in the *gid1* mutant when treated with GA_3_ (Appendix A), indicating that NAL22 acted downstream of GID1 and participated in signal transduction, being negatively regulated by GA.

### 3.3. NAL22 Encoded a Plant-Specific Conserved Protein

The NAL22 in rice is annotated as a multicopy-associated filamentation (Maf)-like protein. Previous research has found that overexpression of the bacterial Maf-like protein of *Bacillus subtilis* resulted in extensive filamentation of cells [47]. Through sequence alignment, 78 homologous proteins with more than 40% similarity to NAL22 were identified in various plant genomes. However, no high-similarity homologs were found in bacteria, yeast, or animals (Appendix A), demonstrating that NAL22 was a plant-conserved protein. The NAL22 protein and its homolog in rice had a unique C-terminal domain not present in other plant species (Appendix A), suggesting that NAL22 may have undergone mutations and artificial selection during rice domestication. In bacteria, Maf-like proteins have nucleotide pyrophosphatase activity and are divided into two subfamilies according to different substrates, and their overexpression inhibits cell division [48]. In contrast, knocking out *NAL22* in rice inhibited cell division in rice. Sequence alignment results showed that the active sites of Maf-domain proteins in bacteria and rice are conserved (Appendix A), but NAL22 did not completely align with the marker sites of any known subfamily (Appendix A), indicating that it belonged to a new subfamily (Appendix A). The predicted protein structures of NAL22 and YhdE, a bacteria Maf-like protein that has a function opposite to NAL22, are significantly different, with NAL22 lacking an enzyme pocket (Appendix A). These differences might explain why NAL22 has an opposite function on cell division compared to YhdE.

## 4. Materials and Methods

### 4.1. Plant Materials

The RDP-II population and SNPs were publicly available [49]. The 351 rice accessions included 146 IND, 52 TEJ, 61 TRJ, 53 ARO, and 39 AUS. These accessions were planted at Tianjin Modern Agricultural Science and Technology Innovation Base (Dannangong Village, Dongpuwa Township, Wuqing District, Tianjin, 117.044 east longitude and 39.384 north latitude) with a loam temperate monsoon climate, sandy and normal field management. *nal22* mutants were generated by CRISPR/Cas9 technology via agrobacterium-mediated transformation on the ZH11 background. The NJ6 and its near-isogenic line *gid1* mutant were provided by Dr. Xiangdong Fu (Institute of Genetics and Developmental Biology, Innovation Academy for Seed Design, Chinese Academy of Sciences, Beijing, China).

### 4.2. Genome-Wide Association Analysis

Leaf width was measured at the widest part of the leaf using a ruler with millimeter precision. Each variety has at least two sets of data. A GWAS was performed following the method described previously [50]. In brief, 131,332 SNPs with minor-allele frequency (MAF) ≥ 0.05 were used for GWAS. The population structure (Q) was calculated using Structure (v2.3.4) software (Stanford University, San Francisco, CA, USA). The Tassel (v4.0) software (Cornell University, Ithaca, New York, NY, USA) was used and a mixed linear model (MLM) was selected to combine the population structure with the genetic kinship matrix (K) generated from the SNP data to perform the GWAS. Perl scripts (version 5.24.1, San Francisco, CA, USA) were then used to draw the integrated Manhattan plot from the GWAS output. Candidate intervals were screened based on a *p*-value less than 1 × 10^−4^.

### 4.3. DNA Isolation and Candidate Gene Sequencing

DNA was isolated from one-month-old plants by the CTAB method. The reference sequences of seven candidate genes were obtained from the MSU rice sequences database (accessed on 3 April 2021) [51]. Primers for amplifying the full-length sequence were designed on NCBI (accessed on 3 April 2021) [52] and listed in Appendix A. DNA sequencing was done at Tsingke Biotechnology Co., Ltd (Beijing, China). Sequence alignment was performed with MEGA5 (Tokyo Metropolitan University, Hachioji, Tokyo, Japan).

### 4.4. Expression Pattern Analysis

The expression patterns of genes in LALWs were analyzed on Plant Regulomics (accessed on 8 September 2021) [53]. Firstly, the species was selected as “Oryza sativa”, and then the RAP-DB IDs of genes were inputted. Finally, the “Tissue Expression Profile” was selected for analysis.

### 4.5. Histology Observation

Leaves for histology observation were taken from the second leaf of 85-day-old plants. The tissue sections were prepared from the widest part of the leaf and removed by hand with a surgical blade. The number and width of veins and width of seeds were observed and measured with a stereomicroscope (SZX-16 OLYMPUS, Japan). Each sample had at least 10 replicates for observation.

### 4.6. RNA Isolation and Quantitative Real-Time Polymerase Chain Reaction (qRT-PCR)

Samples for RNA isolation were collected from different plants or tissues, frozen immediately with liquid nitrogen, and ground into powder in a mortar in liquid nitrogen. RNA was extracted using a kit from Sangon Biotech (B511321, Shanghai, China). The kit used for reverse transcription was from Vazyme (R312, Nanjing, China). Primers for qRT-PCR were designed on qPrimerDB (accessed on 15 November 2021) [54] and listed in Appendix A. *Ubiquitin* was used as the internal control for relative quantification, and the relative transcription level of control was used as the calibrator. qRT-PCR was performed with a ChamQ Universal SYBR qPCR Master Mix (Vazyme, Q711, Nanjing, China) and ran on an ABI QuantStudio™ 6 (Thermo Fisher Scientific Inc., Waltham, MA, USA) machine. The 2^−ΔΔCT^ method was used to calculate relative expression levels [55]. Each reaction was replicated three times.

### 4.7. Phylogenetic Tree Construction and Conserved Motif Prediction

The sequences of homologous proteins were downloaded from the NCBI website (https://www.ncbi.nlm.nih.gov/, accessed on 28 December 2022). Phylogenetic trees were constructed using a neighbor-joining method in MEGA5. MEME (accessed on 3 April 2022) [56] was used for conservative domain prediction. Firstly, we needed to enter the MEME interface in Motif Discovery. The “Classic model” was selected as the motif discovery model. After importing the protein sequence, “Zero or One Occurrence Per Sequence (zoops)” was selected in the site distribution column. The number of motifs expected to be found was set to 50. Finally, we set the motif width from 6 to 100 in “Advanced options”.

### 4.8. GA_3_ and PAC Treatment

GA_3_ (G8040, Solarbo, Beijing, China) or PAC (P8790, Solarbo, Beijing, China) was used to treat rice seedlings at a concentration of 100 μmol L^−1^. In brief, two-week-old seedlings were used for the following research, and the compound was sprayed evenly on the leaf surface using a spray bottle to form a water mist, and then sealed with plastic wrap to moisturize. Each treatment was repeated three times.

### 4.9. cis-Acting Element Prediction

Prediction of *cis*-acting elements in the promoter of *NAL22* was performed using online tools of NEW PLACE (accessed on 1 June 2022) [57] and PLANTCARE (accessed on 1 June 2022) [58].

### 4.10. Accession Numbers

We submitted the *NAL22* gene sequence to GenBank and the accession number is ON677251. The other 23 genes used in this study are publicly available and downloaded in GenBank (accessed on 9 September 2022). Their names and accession numbers are: *CG1-CG6* (accession numbers: LOC4333916, LOC4333930, LOC107275925, AL662958, LOC4336479, CR855229), *CDC2OS-1* (LOC4331415), *CDC2OS-2* (LOC4328135), *R2* (LOC4338689), *EXPA10* (LOC4336776), *EXPB5* (LOC4336602), *GH9B1* (LOC4330637), *EC_YhdE* (P25536), *BS_Maf* (Q02169), *PF0216* (Q8U476), *Mbar-A1652* (Q46BZ6), *SC_MaF* (Q99210), *HS_MaF* (O95671), *EC_YceF* (P0A729), *STM_YceF* (P58627), *Dd_Maf* (Q54TC5), *PA2972* (Q9HZN2), *Tb_Maf1* (Q382A9), respectively.

## 5. Conclusions

Leaf width is an important factor in rice breeding, as it can affect the overall photosynthetic area and yield of rice plants. This study presents the discovery of a novel rice gene, *NAL22*, which is related to leaf width and was identified by GWAS and CRISPR/Cas9 gene editing technology. The polymorphisms and expression level of *NAL22* were found to be correlated with the leaf width of different varieties. This gene functioned downstream of *GID1* and was negatively regulated by GA. Results showed that *NAL22* regulated leaf width by affecting vein width and cell division while not affecting seed width. These findings offer a new possibility for rice breeding with the ideal plant type.

## Figures and Tables

**Figure 1 ijms-24-04073-f001:**
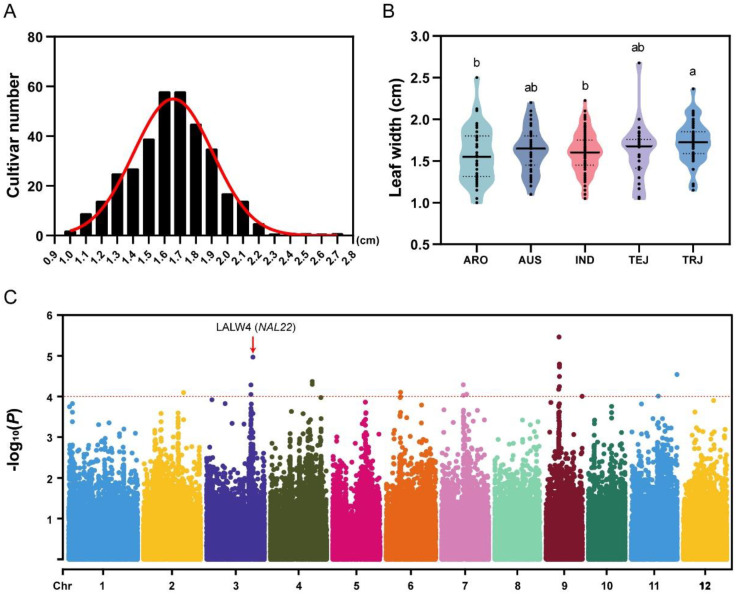
LW variation in the RDP-II population and corresponding GWAS results. (**A**) The LW variation distribution in the RDP-II population. The X-axis represents LWs, and the Y-axis represents the number of accessions with the LW interval. (**B**) LW variation in the subpopulations. Each dot represents a rice accession in the subpopulation. Black lines represent the average value of LWs in the five subpopulations. “a” and “b” represent significant differences (*p* < 0.05) of the leaf width among different subpopulations. (**C**) GWAS of the LW variation using the RDP-II population and 700,000 SNP maps. The X-axis represents 12 rice chromosomes, and the Y-axis represents the converted *p*-value (−log10(P)). The dotted line represents the threshold of candidate SNPs, and the dots above that line are candidates. The dot marked with a red arrow represents the LALW4.

**Figure 2 ijms-24-04073-f002:**
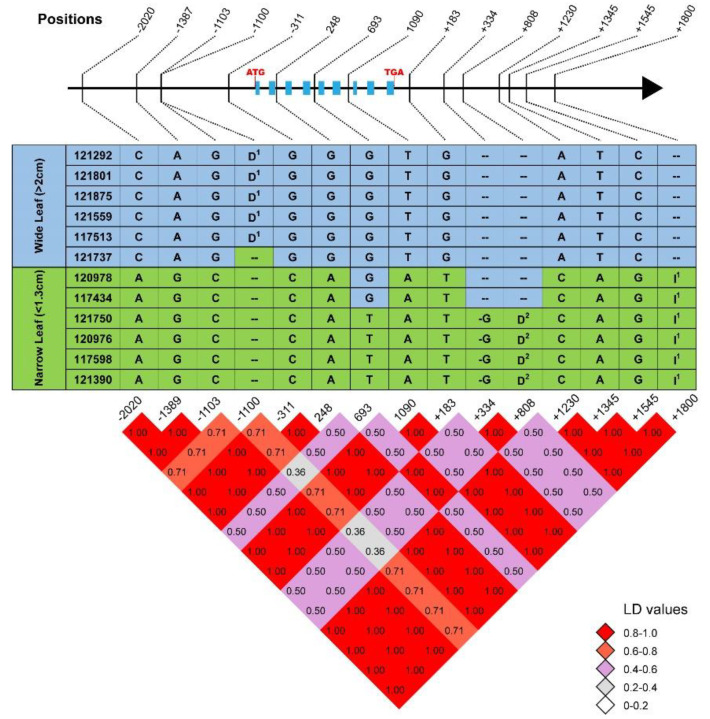
15 polymorphisms of *CG7* were related to rice leaf-width variation. Tilted numbers at the top represent the position: “-” indicates upstream of “ATG”; “+” indicates downstream of “TGA”; the unmarked number indicates exons and introns. Black rectangles indicate the gene exon regions. The middle panel shows the detailed sequence variation among different accessions. “A, T, G and C” in the table are the bases. ‘--’ indicates that the sequence in the accessions is the same as the reference sequence, and “-” indicates deletion. “D^1^” and “D^2^” represent deletion of “TTAACT” and “CCTTGTGCCACTG”, respectively. “I^1^” represents the insertion of “CGCCGCCGCCGC”. The bottom panel is the LD decay of the LW variation in the region. The LD values are included in the blocks with different colors.

**Figure 3 ijms-24-04073-f003:**
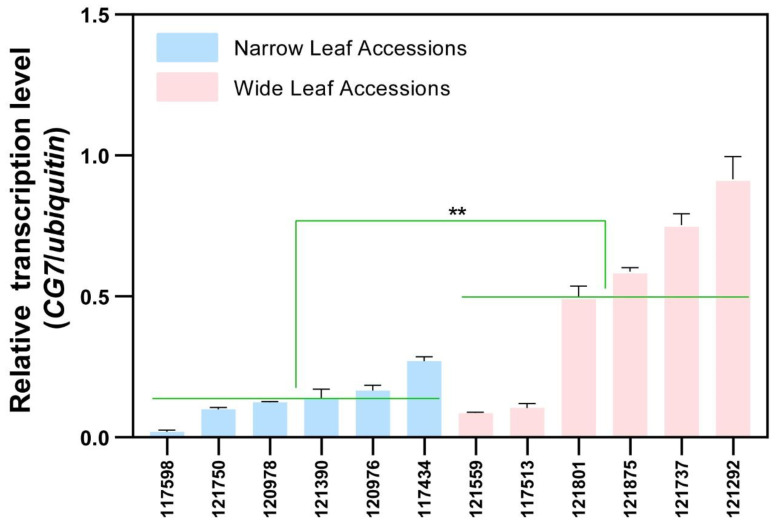
The transcription level of *CG7* was positively associated with rice leaf width. Blue bars indicate accessions with narrow leaves (<1.3 cm); pink bars indicate accessions with wide leaves (>2 cm). “**” represents significant differences (*p* < 0.01) of the transcription level of *CG7* between wide leaf accessions and narrow leaf accessions.

**Figure 4 ijms-24-04073-f004:**
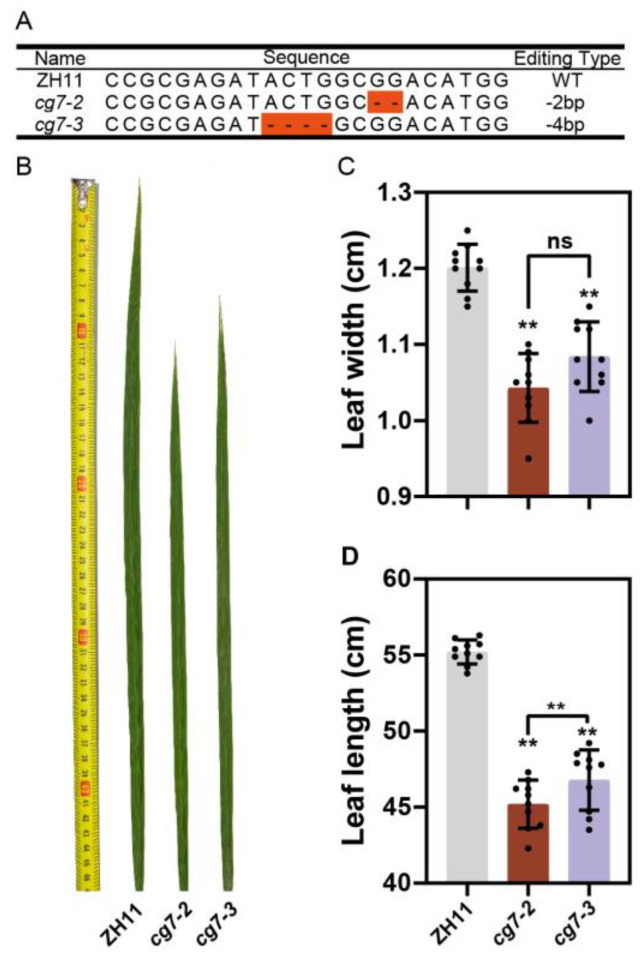
*nal22* mutants had narrower and shorter leaves. (**A**) Editing type of *nal22* mutants. “-2” and “-3” below the name represent mutant numbers. “--” with orange background indicates two base deletions. (**B**) The phenotype of ZH11 and *nal22* mutants. (**C**,**D**) Bar diagram of leaf width and leaf length. Gray bars indicate ZH11; red bars indicate *nal22-2*; purple bars indicate *nal22-3*. Black bars indicate the difference between two *nal22* mutants. “**” represents significant differences (*p* < 0.01), and “ns” represents no significant difference.

**Figure 5 ijms-24-04073-f005:**
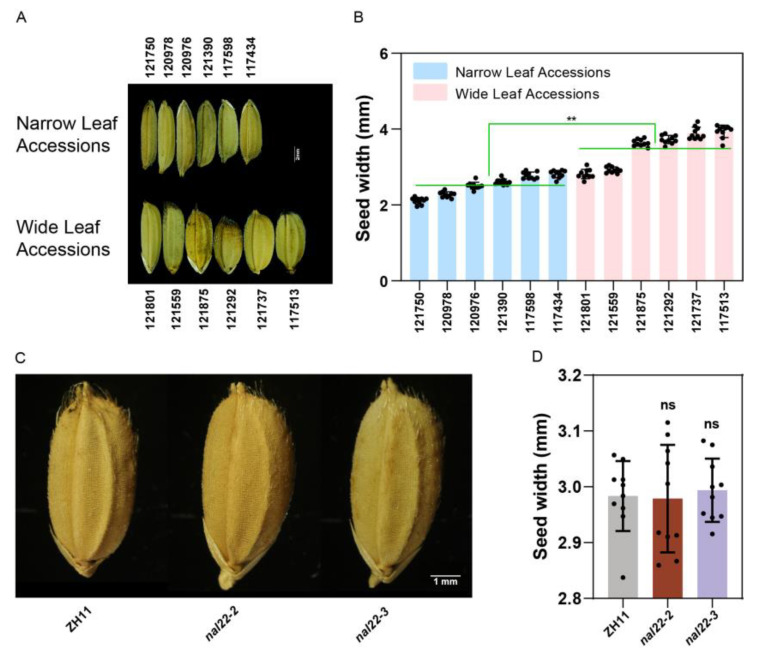
*NAL22* has no effect on seed width. (**A**) The phenotype of seed width of different leaf-width accessions. Bar = 2 mm. (**B**) Bar diagram of seed width. Blue bars indicate accessions with narrow leaves (<1.3 cm); pink bars indicate accessions with wide leaves (>2 cm). (**C**) The phenotype of seed width of ZH11 and *nal22* mutants. Bar = 1 mm. (**D**) Bar diagram of seed width. Gray bars indicate ZH11; red bars indicate *nal22-2*; purple bars indicate *nal22-3*. Black bars indicate the difference between two *nal22* mutants. “**” represents significant differences (*p* < 0.01), and “ns” represents no significant difference.

**Figure 6 ijms-24-04073-f006:**
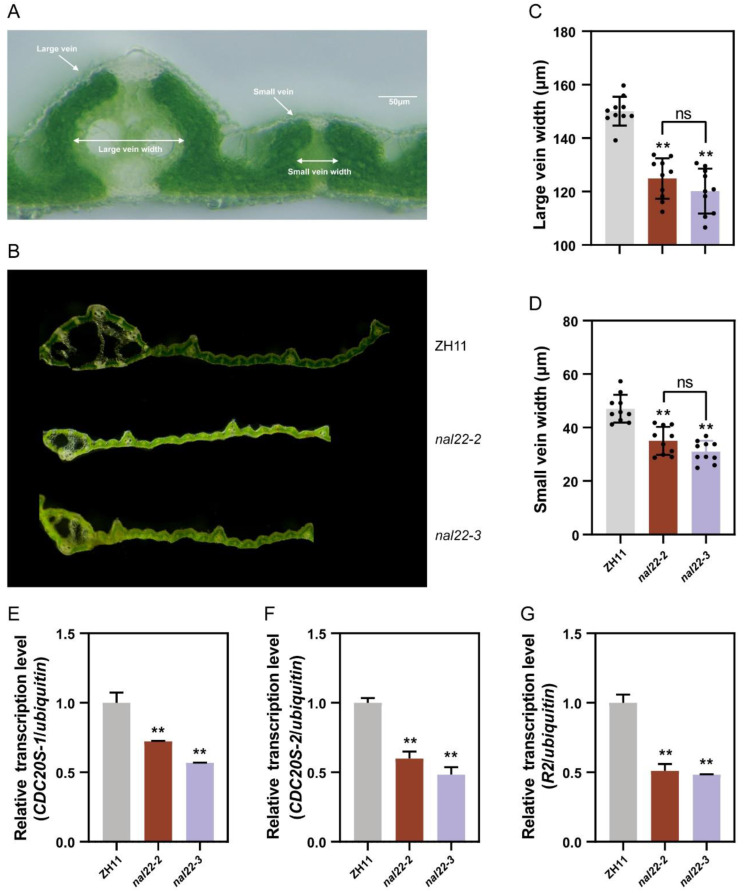
*NAL22* affected the vein width and transcription level of genes related to cell division. (**A**) Schematic and diagram of large veins, small veins, large vein width and small vein width. (**B**) Transverse sections of the mature second leaves of wild-type and *nal22* mutants. Bar = 500 μm. (**C**,**D**) Bar diagram of large vein width and small vein width. (**E**–**G**) Bar diagram of transcription level of *CDC20S-1*, *CDC20S-2*, and *R2*. Gray bars indicate ZH11; red bars indicate *nal22-2*; purple bars indicate *nal22-3*. Black bars indicate the difference between two *nal22* mutants. “**” represents significant differences (*p* < 0.01), and “ns” represents no significant difference.

**Figure 7 ijms-24-04073-f007:**
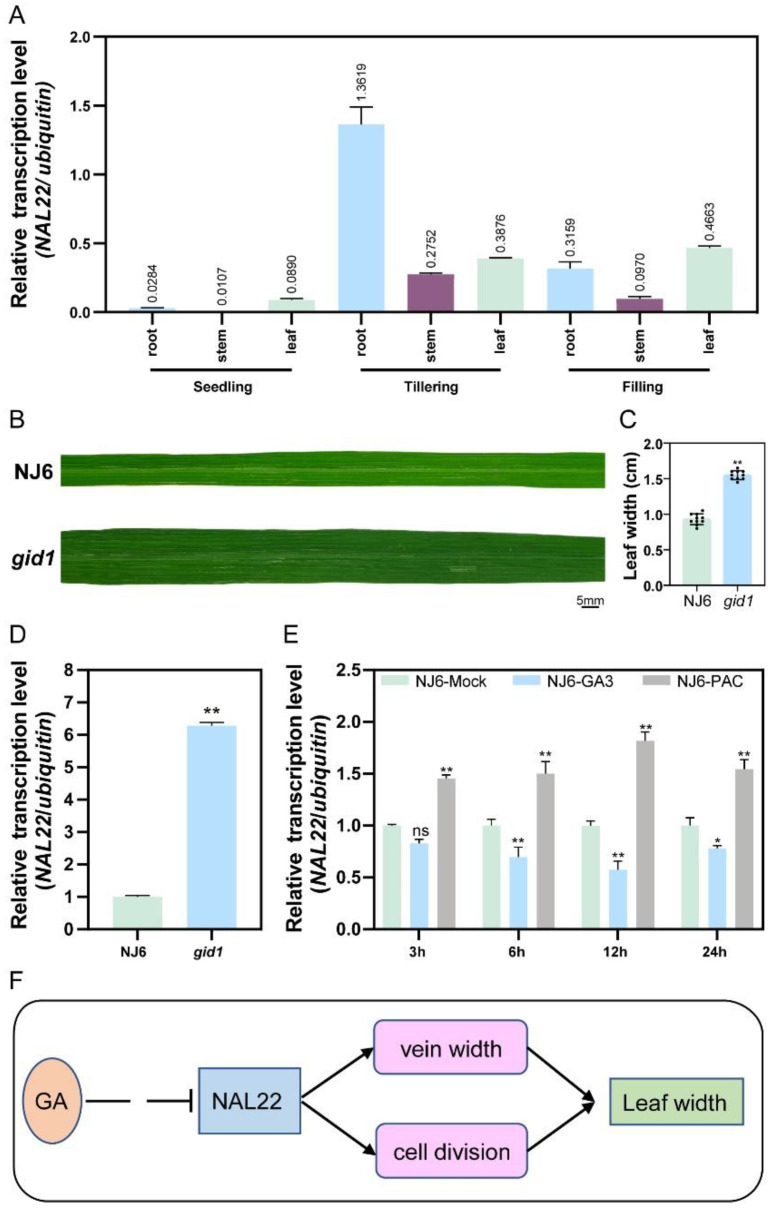
*NAL22* was negatively regulated by GA. (**A**) The transcription level of *NAL22* in different tissues. “Seedling, tillering and filling” indicate different growth stages. The number above the column indicates the transcription level of *NAL22* relative to ubiquitin. (**B**) Leaf-width phenotype of NJ6 (wild type of *gid1*) and *gid1*. Bar = 5 mm. (**C**) Bar diagram of leaf width of NJ6 and *gid1*. (**D**) The transcription level of *NAL22* in NJ6 and the *gid1* mutant. (**E**) Transcription level of *NAL22* under water, GA_3_ and PAC treatment at different time points. (**F**) The proposed model for *NAL22* mediating leaf width in rice. This model shows that *NAL22* promotes rice leaf width by affecting vein width and cell division, while GA suppresses *NAL22*. The concentrations of GA_3_ and PAC were 100 μmol L^−1^. “*” represents significant difference (*p* < 0.05). “**” represents significant difference (*p* < 0.01), and “ns” represents no significant difference.

## Data Availability

The data presented in this study are available on request from the corresponding authors.

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
