# Peer review of "Identification of a Rice Leaf Width Gene Narrow Leaf 22 (NAL22) through Genome-Wide Association Study and Gene Editing Technology"

_ijms, 2023, doi:10.3390/ijms24044073_

Round 1
Reviewer 1 Report
To,
The Editor,
IJMS, MDPI,
Manuscript ID: ijms-2190378
Subject: Submission of comments of the manuscript in “IJMS"
Dear Editor IJMS, MDPI,
Thank you very much for the invitation to consider a potential reviewer for the manuscript (ID: ijms-2190378). My comments responses are furnished below as per each reviewer’s comments.
In the reviewed manuscript, the authors performed a genome-wide association study (GWAS) of rice leaf width with 351 accessions from the rice diversity population â…¡. A total of 12 loci associated with leaf width (LALW) were identified. We identified one gene in LALW4 that was genetically linked to leaf width variations. Its expression level was positively associated with rice leaf width. Through CRISPR/Cas9 gene editing technology, we knocked out this gene in rice, and the leaf of mutants became narrower. We named it Narrow Leaf 22 (NAL22). The vein width and expression level of genes involved in cell division were suppressed in the nal22 mutant. In addition, gibberellin (GA) negatively regulates NAL22. In summary, we identified a rice gene, NAL22, which is related to vein width and cell division, negatively regulated by GA, and controls rice leaf width. Given that rice architecture is one of the most important phenotypes in modern rice breeding, NAL22 has potential application value for leaf shape design. These results provide information for dissecting the genetic and molecular basis of development and improvement through molecular breeding and transgenic approach. In general, the manuscript represents a very big piece of information in a logical presentation. Therefore, it might be conditionally accepted subject to major revision. Authors have to improve their manuscripts with many non-clear meanings, inaccuracies and inconsistencies, and the authors need to address the following issues before it can be accepted for publication.
- I have read the entire manuscript and my initial comment is that manuscript is poorly written. I have significant concerns about the grammar and vocabulary of the manuscript; therefore, I recommend the authors to used an English proofreading service.
- The structure of the abstract should be improved, as well as the lack of several aspects that should be included in this section. The abstract should highlight the most important results of the parameters and characteristics assayed.
- The figures in this section, some figures are quite low resolution and difficult to make out. Higher-resolution versions will be needed for publication. Further, figure texts are not readable, for example, in Figures 2, 5B, 5D, 7A, 7C, and 7E.
- In Material and Methods:- indicate how many replicates assayed in each analysis/parameter. The number of samples or biological and technical replicates should be mentioned for each parameter in the methods.
- Material methods most the citation is the webpage, some website is not working, hence, better to cite the original research paper.
- Results must be explained clearly and in detail.
- qRT-PCR methodology provided is also very vague and confusing. Please provide more details like what was the calibrator used in the study. I assume the authors have used the control as the calibrator. If so, the authors should not include the control within the bar graph as it represents the fold change between the treated vs control and a fold change of “1” for the ‘control’ doesn’t make any sense. Also, would be good to provide details on what reagents (details of probes used, if any, if SYBR was used then details for that, etc.) and real time PCR machine were used in the current study.
- The discussion should be interpreted with the results as well as discussed in relation to the present literature.
- The conclusion section is very poorly written. It should be extensively improved.
- References: shall have to correct the whole References according to the ”Instructions for the Authors”, e.g. the Journal name and scientific name must be italics, the year must be bold and you shall have to use the abbreviated name of the Journals cited, authors must specify journal name. Authors just mention the Journal article in name of the journal in all references. The authors must correct it.
Reviewer 2 Report
The study on Identification of a rice leaf width gene Narrow Leaf 22 (NAL22) through genome wide association study and gene editing technology is worthy of investigation and reporting, and information would be useful to understand the rice architecture. However, there are major corrections and supporting data is lacking to validate the current study and conclude the role of NAL22 gene in optimizing plant architecture and ultimately increase grain yield in high yielding rice cultivars. The comments are appended below:
1- Manuscript is lacking agronomic data to understand how NAL 22 gene is affecting the plant height, days to maturity, panicle length , number of grains/ per panicle and total grain yield. Authors could have provided these information’s in the main Table of manuscript to conclude the importance of gene in developing high yielding cultivars/hybrids.
2- Authors found NAL22 has no effect on seed width in the present study, what about grain length(L), width(W) and L/W ratio and grain weight . Is any correlation found with these traits?
3- Is authors found any distribution pattern of NAL 22 gene (narrow leaf) among the 351 rice accessions used in this study taken as RDP(II)? It will be interesting to see among the subgroups of rice(indica, tropical and temperate japonica, aus and aromatic) how leaf width trait has been distributed, Is any correlation found. Even, authors can mention these information’s in discussion part.
4- Conclusion of the present study is lacking. Authors should add conclusion section mentioning the role of NAL 22 gene in optimizing the rice architecture and how this candidate gene will be useful for breeding super rice cultivars with increased yield potential and superior quality
Reviewer 3 Report
Authors should review the manuscript to eliminate some errors that must be corrected before the publication of the manuscript.
Author should revise the manuscript according to the guide for authors.
The citation in the text should be edited.
A different type of fonts and size of font appears in the manuscript: eg., lines 43-45 and 332. There are more cases.
Abbreviations (e.g., RDP) should be defined the first time they appear.
SI units should be used.
The titles of the figures should be concise.
Some information is repeatedly presented in the manuscript, so authors should accurately check the logical flow of the information in the manuscript.
The list of references should be accurately reviewed according to the guidelines.
Reviewer 4 Report
1. The abstract does not report the main findings of the study in a clear manner. For example general expressions are used which do not provide useful information to the readers. Information that is more specific is required in the abstract.
2. The introduction does not point out the gap of the literature the study seeks to fill and novelty of the study over the existing literature. This point showed be further elaborated.
3. Key words must be arranged alphabetically
4. The objectives and conclusion of the study are not clear and need to re-write
5. In Materials and Methods section more information is required (Plant materials) e.g soil type and its properties. Irrigation and crop management. Climate conditions
6. doi should always be added when available
Round 2
Reviewer 1 Report
Dear Editor,
Thank you for providing the opportunity to review the revised manuscript. The manuscript is improved considerably after revision according to the reviewer's comment. Now this study is a suitable contribution to the IJMS. I recommend the manuscript for publication.
Thank you
With best regards
Reviewer 2 Report
I agree with the authors justification/ revisions.
Reviewer 4 Report
Dear Editor Journal of international journal of molecular science
Manuscript ID: ijms-2190378
I re-reviewed the manuscript “Identification of a rice leaf width gene Narrow Leaf 22 (NAL22) through genome wide association study and gene editing technology” again and the authors made all the amendments that I asked before so I think the manuscript is suitable for publishing
Regards